# Evidence for saddle point-driven charge density wave on the surface of heavily hole-doped iron arsenide superconductors

Quanxin Hu[1,7], Yu Zheng[1,7], Hanxiang Xu[2], Junze Deng [2,3], Chenhao Liang[2,3], Fazhi Yang[1], Zhijun Wang [2,3], Vadim Grinenko[1], Baiqing Lv[1,4] ✉, Hong Ding[1,5,6] ✉ & Chi Ming Yim [1] ✉

Unconventional superconductivity is known for its intertwining with other correlated states, making exploration of the intertwined orders important for understanding its pairing mechanism. In particular, spin and nematic orders are widely observed in iron-based superconductors; however, the presence of charge order is uncommon. Using scanning tunnelling microscopy, and through expanding the phase diagram of iron-arsenide superconductor $Ba_{1-x}K_xFe_2As_2$ to the hole-doping regime beyond $KFe_2As_2$ by surface doping, we demonstrate the formation of a charge density wave (CDW) on the arsenide surface of heavily hole-doped $Ba_{1-x}K_xFe_2As_2$. Its emergence suppresses superconductivity completely, indicating their direct competition. Notably, the CDW emerges when the saddle points approach the Fermi level, where its wave-vector matches with those linking the saddle points, suggesting saddle-point nesting as its most probable formation mechanism. Our findings offer insights into superconductivity and intertwined orders, and a platform for studying them in iron-based superconductors close to the half-filled configuration.

The phase diagrams of high-temperature superconductors including cuprates and iron pnictides are typically complex, with multiple types of correlated orders (such as nematic[1], spin[2], and charge order[3,4]) occurring at similar energy scales. These coexisting/competing electronic ordered states have been widely studied due to the proposition that their fluctuations may play a role in facilitating superconducting pairing. Among them, charge density wave (CDW) orders have attracted extensive attention. It is now evident that the widespread occurrence of CDW order is intricately linked with the phenomenon of superconductivity in cuprates, and CDW correlations play a crucial role in shaping the cuprate phase diagram[5–9]. In contrast to cuprates, where the principal characteristics can be encapsulated by a one-band Hubbard model, all $d$ orbitals of Fe atoms play an important role in

iron-based superconductors (FeSCs). This leads to insights into the unusual metallic states governed by Hund's coupling[10]. As a result, the ubiquitous intertwined order of FeSCs takes form in a symmetry-breaking nematic phase and a spin density wave (SDW). Although some charge modulations have been reported in FeSCs[11–14], associated evidences including clear contrast inversion between STM images at opposite polarities and the associated opening of an energy gap are rarely reported. The interplay between CDW and superconductivity in FeSCs becomes a long-standing puzzle, which is seldom addressed in the current phase diagram.

To obtain a comprehensive understanding of the intertwined orders in FeSCs, it is imperative to investigate such a theme in a much wider hole-doping range. 122-type FeSCs, where the $Fe_2As_2$ layers are

[1]Tsung-Dao Lee Institute & School of Physics and Astronomy, Shanghai Jiao Tong University, Shanghai, China. [2]Beijing National Laboratory for Condensed Matter Physics, Institute of Physics, Chinese Academy of Sciences, Beijing, China. [3]School of Physical Sciences, University of Chinese Academy of Sciences, Beijing, China. [4]Zhangjiang Institute for Advanced Study, Shanghai Jiao Tong University, Shanghai, China. [5]Hefei National Laboratory, Hefei, China. [6]New Cornerstone Science Laboratory, Shanghai, China. [7]These authors contributed equally: Quanxin Hu, Yu Zheng. ✉e-mail: baiqing@sjtu.edu.cn; dingh@sjtu.edu.cn; c.m.yim@sjtu.edu.cn

interleaved with Ba or other alkali elements, offer exceptional suitability for investigating the intertwined orders and their interactions, due to their high-quality single crystals, easily cleaved surfaces, and flexible chemical substitutions. Pure $BaFe_2As_2$, characterized by six electrons per Fe atom (6 electrons/Fe), becomes structurally distorted and magnetic at temperatures below ~140 K. Further hole-doping leads to the emergence of superconductivity accompanied by suppression of the magnetic phases. The superconducting order parameter of lightly hole-doped 122-type FeSCs $Ba_{1-x}K_xFe_2As_2$ is most likely to be $s\pm$, with opposite gap signs between the hole and electron pockets, but with the same sign between the two central hole pockets[15]. The hole doping can continue until reaching another stoichiometric compound $KFe_2As_2$ (characterized by 5.5 electrons/Fe), in which the electron pockets vanish and additional propeller-shaped hole pockets appear at the Brillouin zone corners[16]. The superconducting transition temperature ($T_c$) of $KFe_2As_2$ is about 3 K, an order of magnitude less than that of the optimally doped samples. Both the $d$- and $s$-wave pairing order parameters are predicted to be favourable in $KFe_2As_2$, and which state becomes dominant depends on the ratio between the inter-band electron/electron-pocket and electron/hole-pocket interactions[17–22]. Experimentally, the measurements of thermal conductivity[23,24], specific heat[25], and penetration depth[26] point towards the $d$-wave gap symmetry, while a nodal $s$-wave gap is favoured by laser ARPES measurements[27,28], making the pairing order parameter and the location of the gap nodal point in $KFe_2As_2$ remain elusive. Theories predict that as the $d^5$ configuration is reached, a Mott insulating state is formed. This Mott insulating state is strongly favoured by Hund's coupling and may influence a large regime of the phase diagram[29,30]. This suggests the presence of other correlated states at higher levels of hole doping, indicating that the superconductivity observed in $KFe_2As_2$ may arise from these ordered states. Unfortunately, $KFe_2As_2$, which has 5.5 electrons/Fe, represents the most extended doping level that can be continuously achieved in a single family of FeSCs to date.

The polar nature of the cleaved surfaces of 122-type FeSCs along with the accompanying surface doping effect offer a captivating platform for studying the material in an extended range of hole doping levels. Here, we systematically studied the As-terminated surface of 122 compound $Ba_{1-x}K_xFe_2As_2$ with various doping $x$ ($x \approx 0.45, 0.77, 1$) using low-temperature scanning tunnelling microscopy (STM). Both STM measurements and density functional theory (DFT) calculations confirm the effective hole doping levels on the topmost As surface layer of $Ba_{1-x}K_xFe_2As_2$. As $x$ increases, we observe the emergence of a static, checkerboard-like charge density wave (CDW) order with a spatial periodicity of $2a_{As} \times 2a_{As}$ on the As-terminated surface of $Ba_{1-x}K_xFe_2As_2$ ($x \approx 0.77$ and $x = 1$), on which features associated with superconductivity are completely suppressed. The CDW wavevector matches well with the nesting vector between the saddle points located very near the Fermi level ($E_F$), providing convincing evidence for the saddle-point-nesting scenario for the CDW formation. Also, our work expands the phase diagram of 122-type FeSCs and highlights the intimate connection between superconductivity and charge order in heavily hole doped iron arsenide superconductors.

## Results

### Two different surface terminations of $Ba_{1-x}K_xFe_2As_2$ ($x \approx 0.77$)

The crystal structure of 122-type FeSCs (Fig. 1a) comprises layers of spacer atoms Ba/K between the $Fe_2As_2$ layers. As illustrated in Fig. 1b, cleavage of $Ba_{1-x}K_xFe_2As_2$ samples can result in two different terminated surfaces: the Ba/K-terminated surface and the As-terminated

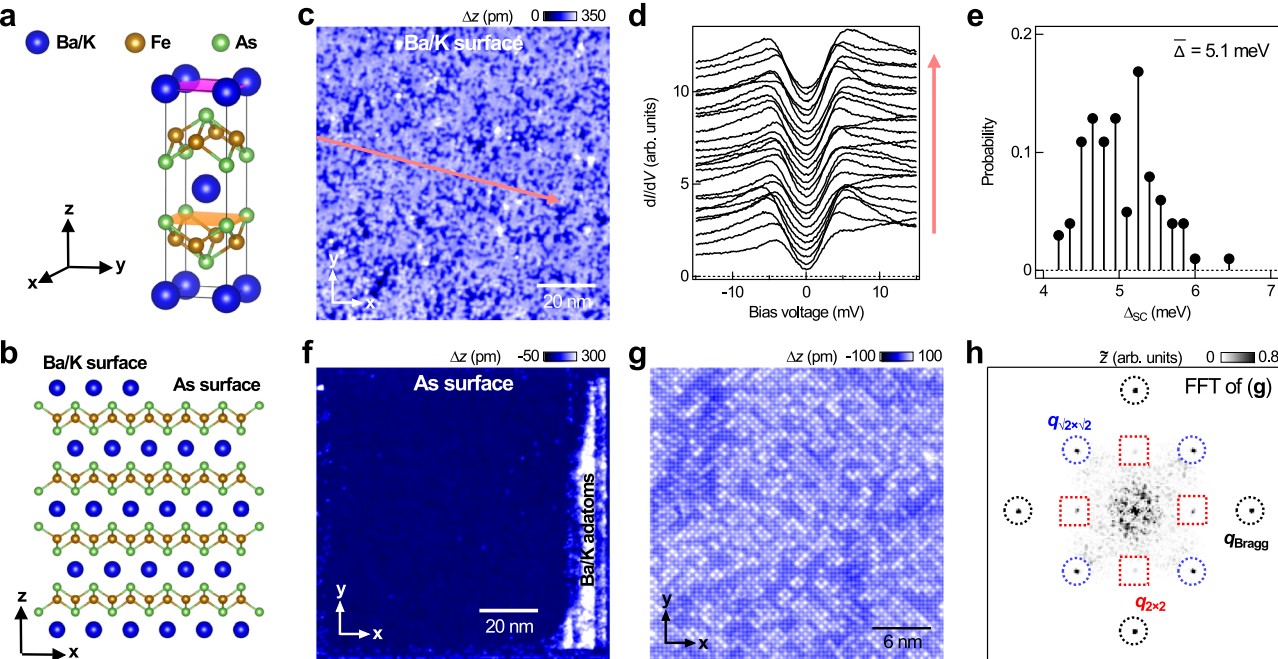

**Fig. 1 | Identification of surface terminations of $Ba_{1-x}K_xFe_2As_2$ ($x \approx 0.77$).**
**a** Crystal structure of $Ba_{1-x}K_xFe_2As_2$ ($x \approx 0.77$). **b** Its two possible cleaved surfaces: Ba/K- and As- terminated surfaces (side view). **c** STM topographic image obtained from the disordered Ba/K terminated surface [($V, I$) = (80 mV, 80 pA), image size: (100 nm)²]. **d** Differential conductance (d$I$/d$V$) as a function of bias voltage ($V$) and position ($x$) spectra measured along the red arrowed line in **c** [spectroscopic setpoint ($V_s, I_s$) = 15 mV, 200 pA; amplitude of bias modulation used $V_{mod}$ = 0.25 mV]. Spectra were obtained at a temperature of 4.2 K. Spectra are vertically offset for clarity. **e** Histogram of the superconducting gap sizes ($\Delta_{SC}$), calculated by measuring the separation distance between the coherence peaks. **f** Large-scale topographic image of As-terminated surface obtained after the displacement of the Ba/K atoms to the edges of the imaged region [($V, I$) = (80 mV, 100 pA), image size: (110 nm)²]. **g** Atomically-resolved topographic image obtained from the central part of **f** [($V, I$) = (− 10 mV, 300 pA), image size: (40 nm)²]. The image shows that the As-terminated surface exhibits (1 × 1) As- unit cell, and ($\sqrt{2} \times \sqrt{2}$) spatial periodicity originating from surface reconstruction. **h** Fast Fourier transform of **g**, where black circles mark the (1 × 1) unit cell Bragg peaks, blue circles mark the peaks associated with the ($\sqrt{2} \times \sqrt{2}$) surface reconstruction, and red squares mark the additional ordering peaks.

surface. Figure 1 shows these surface terminations of the $Ba_{1-x}K_xFe_2As_2$ ($x \approx 0.77$) single crystal sample. The disordered surface (Fig. 1c) exhibits inhomogeneous superconductivity with superconducting gap sizes ($\Delta_{SC}$) ranging from 4.2 to 6.5 meV (Fig. 1d, e). In previous work, the disordered surface was assigned to the Ba/K-terminated surface, with its disorderedness likely caused by the mixture of Ba and K atoms[31]. The inhomogeneity of the superconducting gap sizes arises from chemical disorder across the surface. Our data agree well with previous scanning tunnelling microscopy/spectroscopy (STM/S) studies[32,33].

Apart from the disordered surface, we also observed a distinct atomically flat surface itself covered with randomly scattered protrusions. The protrusions can be easily displaced by scanning with the STM tip brought to a very close distance to the sample (Fig. 1f), indicating their weak bonding to the surface (also see Supplementary Figs. S1 and S2). Furthermore, once displaced, the protrusions tend to aggregate to form a new surface, with a measured height consistent with the $\sqrt{2} \times \sqrt{2}$ reconstruction of the K-terminated surface (see Supplementary Fig. S2 for more details). Turning our focus back to the image taken from the surface with all the protrusions removed by the STM tip, unlike the Ba/K atoms on the Ba/K terminated surface, the atoms on this surface exhibit more pronounced undulations, facilitating clearer atomic resolution at lower set-point currents. On this basis, we assign this termination as the As-terminated surface (see Supplementary Note 1 for more details). The image (Fig. 1g) shows that the As-terminated surface comprises randomly distributed bright and dark regions that exhibit different symmetries with respect to the ($1 \times 1$) As lattice. In the Fourier transformation (Fig. 1h), in addition to the ($1 \times 1$) lattice peaks (marked with black circles), two other sets of peaks, one with ($\sqrt{2} \times \sqrt{2}$) (marked by blue circles) and another with ($2 \times 2$) (marked by red squares) spatial periodicity, are also present. Similar peaks were also observed on the As-terminated surface of $KFe_2As_2$, but not on that of $Ba_{1-x}K_xFe_2As_2$($x \approx 0.45$) (Supplementary Fig. S6). The ($\sqrt{2} \times \sqrt{2}$) pattern may be attributed to the magnetic order present within the top $Fe_2As_2$ layer of $Ba_{1-x}K_xFe_2As_2$, which can be effectively tuned by the ratio of the Ba/K atoms[34]. In the following, we focus mainly on the ($2 \times 2$) order formed along the As-As directions, which have never been reported.

## Spectroscopic characteristics of the (2 × 2) order

To better visualise the ($2 \times 2$) modulation along the As-As directions, we performed tunnelling spectroscopy measurements on the As-terminated surface. To minimize the set-point effect, we present our spectroscopic map data as normalised conductance, $L(\mathbf{r}, V) = [dI/dV(\mathbf{r}, V)]/[I(\mathbf{r}, V)/V]$ rather than the more commonly used differential conductance $dI/dV(\mathbf{r}, V)$[35]. The left panel of Fig. 2b shows the $L(x, V)$ line-cut taken along the red arrowed line in Fig. 2a, together with its second derivative with respect to position plot shown in the middle panel, which shows more clearly the out-of-phase shift of the conductance signals at opposite bias voltages. To further illustrate such phase shift in intensity modulations, from the $L(x, V)$ plot we extracted two line profiles, one at bias voltage of $-15$ mV and another at $+15$ mV, with each of them subtracted by its respective spatial average. As shown in the right panel of Fig. 2b, the two line profiles are perfectly out-of-phase with each other. Such contrast reversal is also evident in the $L(\mathbf{r}, V)$ map slices taken at opposite bias voltages (see the complete set of $dI/dV(\mathbf{r}, V)$ map data and the corresponding reciprocal space data in Supplementary Figs. S3 and S4). As shown in Fig. 2c, d, despite the added complexity of the modulation patterns, it is clear that as bias voltage changes from $-4$ (Fig. 2c) to $+4$ mV (Fig. 2d), the $L(\mathbf{r}, V)$ signals along the dashed lines change from their maximum to minimum values.

In addition to the $L(\mathbf{r}, V)$ map slices (Fig. 2c, d), their corresponding reciprocal space map slices (Fig. 2e, f) also evidence the

presence of the ($2 \times 2$) modulation (with the associated peaks marked by red squares). As shown by the energy-resolved Fourier transformation map along the $\Gamma - M$ direction in Fig. 2g, the intensities of the ($2 \times 2$) modulation peaks vary, but their positions stay unchanged with energy. The contrast reversal of the signal modulation at opposite energies and the non-dispersiveness of the modulation peaks in FFT altogether indicate the presence of a CDW order. To illustrate the commensurability of the CDW order with the underlying ($1 \times 1$) As lattice, in Fig. 2h we lay a ball model of the $Fe_2As_2$ surface layer (atop As and Fe atoms) above the $dI/dV(\mathbf{r}, V)$ map slice at $V = -10$ mV. The checkerboard-like pattern formed as a result of the CDW formation is clearly observed in our STM/S measurements.

## Temperature dependent evolution of the CDW

To provide additional evidence for a commensurate CDW order formed on the As-terminated surface of $Ba_{1-x}K_xFe_2As_2$ ($x \approx 0.77$), we performed temperature-dependent measurements on the observed charge modulation. Figure 3a shows representative constant-current differential conductance maps $dI/dV_{cc}(\mathbf{r}, V)$ at $V = -10$ mV measured at temperatures between 1.8 and 10 K, together with their Fourier transformations shown in Fig. 3b. By comparison of the $dI/dV_{cc}(\mathbf{r}, V)$ map data in Fig. 3a with the $dI/dV(\mathbf{r}, V)$ map slices collected via conventional means (Supplementary Fig. S5), we show that the $dI/dV_{cc}(\mathbf{r}, V)$ map data obtained using the constant-current spectroscopic imaging mode, where the current feedback loop was in closed condition, do not contain any artifacts, therefore reveal the same features as the $dI/dV(\mathbf{r}, V)$ map data do. Through comparison of the data obtained at different temperatures, one can see that as the temperature increases (see Fig. 3c), the charge modulation (and the associated peaks in reciprocal space) decreases in intensity and disappears at about 9 K, which we tentatively refer to as the transition temperature of the observed CDW order ($T_{CDW}$).

The formation of a CDW order usually involves electronic states located near $E_F$ and in most cases leads to opening of an energy gap around $E_F$[36–39]. To verify this, we performed tunnelling spectroscopy measurements at a defect-free position on the charge modulation in $Ba_{1-x}K_xFe_2As_2$ ($x \approx 0.77$) at temperatures below and above $T_{CDW}$. Shown in Fig. 3d, the point spectrum obtained at temperature of 300 mK (red curve) exhibits a gap-like feature extending between $-1.4$ and $+2.4$ meV in energy, which disappears above $T_{CDW}$ (black curve). Also, through numerical analysis to the tunnelling spectroscopy data recorded from the charge modulation at the same energy scale, we find that the CDW gap, which comprises a pair of peaks located at energies below ($\Delta_L$) and above $E_F$ ($\Delta_R$) respectively, exhibits inhomogeneous spatial distribution (see Fig. 3e). This could be due to the inhomogeneous distribution of the Ba and K atoms in the Ba/K layer located underneath the $Fe_2As_2$ surface layer.

The observed contrast reversal (Fig. 2), opening of an energy gap and disappearance above a characteristic temperature ($T_{CDW}$) (Fig. 3) altogether form compelling evidence for the existence of a CDW order on the As-terminated surface of $Ba_{1-x}K_xFe_2As_2$ ($x \approx 0.77$). However, considering the bulk electronic band structure of $Ba_{1-x}K_xFe_2As_2$ ($x \approx 0.77$)[40], the wave-vector associated with the checkerboard charge order observed here does not match with any nesting vector formed between any pair of Fermi surface pockets. Meanwhile, no notable anomalies were observed in the transport measurements[41,42]. On this basis, we exclude the possibility of the presence of CDW in the bulk. We also performed measurements on the As-terminated surfaces of $Ba_{1-x}K_xFe_2As_2$ with different $x$ values, from which we found that the CDW persists on the As-terminated surface of $KFe_2As_2$ but not on that of $Ba_{1-x}K_xFe_2As_2$ ($x \approx 0.45$) (see additional data in Supplementary Information). Overall, the observed CDW appears exclusively on the surface, and its presence is influenced by the level of hole doping in the sample. To understand its formation at the As-terminated surface, we need to search for other explanation(s).

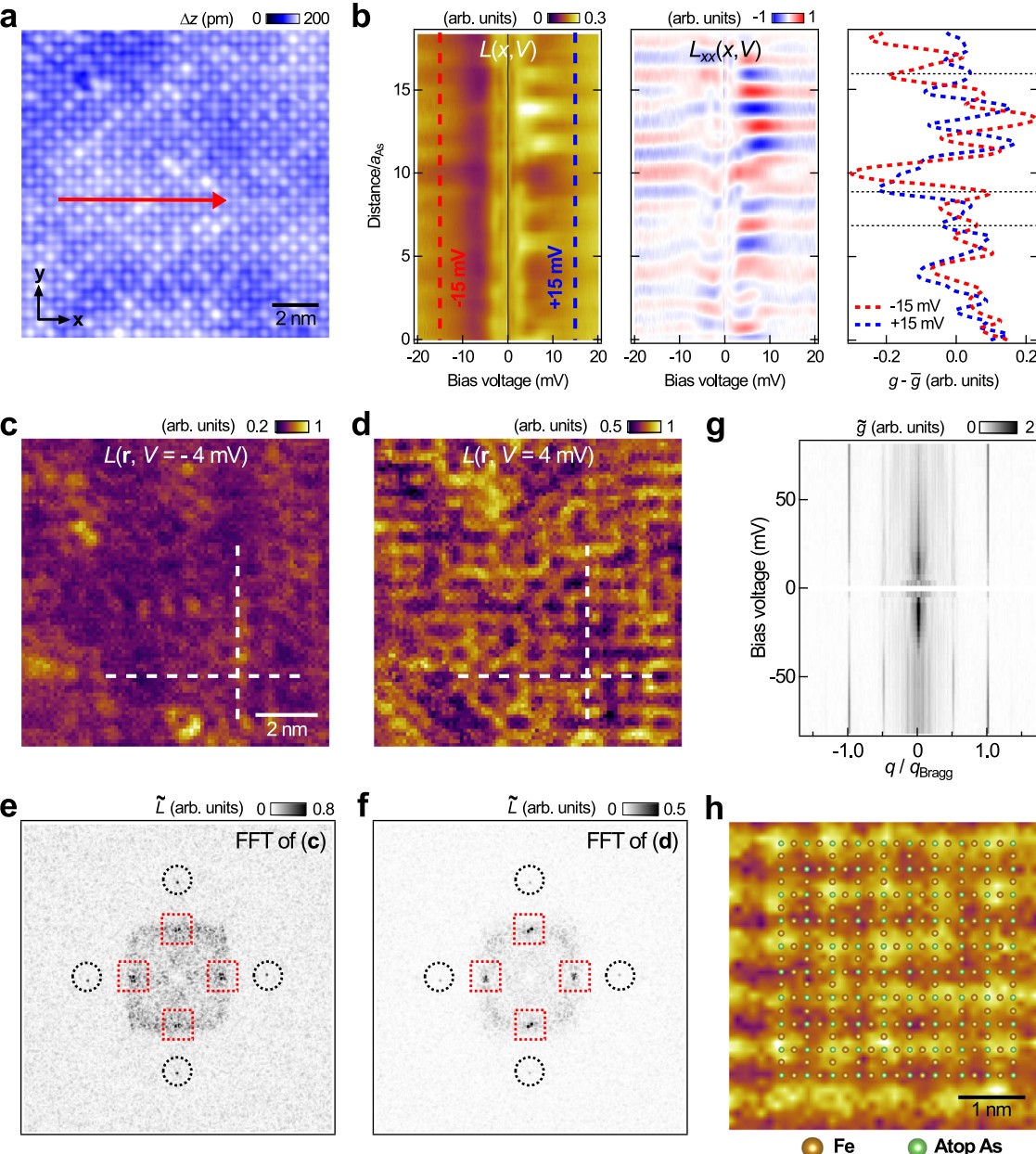

**Fig. 2 | Spectroscopic imaging characterization of the (2 × 2) charge modulation in Ba$_{1-x}$K$_x$Fe$_2$As$_2$ ($x ≈ 0.77$). a** Topographic image of the As-terminated surface of Ba$_{0.23}$K$_{0.77}$Fe$_2$As$_2$ [$(V, I) = (5\,mV, 200\,pA)$, image size: (13 nm)$^2$]. **b** (Left panel) Normalised differential conductance $L(x, V) = (dI/dV)/(I/V)$ as a function of bias and position spectra taken along the red line in **a**, (middle panel) its second-derivative with respect to position, and (right panel) the $L(x)$ plots at bias voltages of $-15$ (red) and $+15$ mV (blue) extracted from the left panel, with each subtracted by the respective spatial average. **c, d** $L(\mathbf{r}, V)$ map slices obtained from a (10 nm)$^2$ area on the As-terminated surface at energies of (**c**) $-4$ and (**d**) $+4$ mV respectively [$(V_s, I_s) = -10\,mV, 200\,pA; V_{mod} = 0.25\,mV$]. Dashed lines mark the same locations between the two map slices. **e, f** The associated Fourier transformations. Black circles highlight the Bragg peaks, red rectangles indicate the (2 × 2) charge order. **g** $\bar{g}(q_x, V)$ along $\Gamma - M$ plot in bias voltage range between $-80$ and $+80$ mV showing the non-dispersive behaviour of the charge order peaks. **h** Zoomed-in of the $L(\mathbf{r}, V)$ map slice recorded at $-10$ mV, overlaid with a ball model of the Fe$_2$As$_2$ surface layer.

## Formation mechanism of the CDW

In the case of 122-type FeSCs, Ba/K atoms act as electron donors, overall providing equal number of electrons to the adjacent Fe$_2$As$_2$ layers. However, when an As-terminated surface is exposed to the vacuum, the Ba/K layer originally above the Fe$_2$As$_2$ surface layer becomes absent, as a result only the Ba/K layer underneath donates electrons to the Fe$_2$As$_2$ surface layer, leading to effective hole-doping to the top surface layer. To understand qualitatively the influence of surface hole-doping on the electronic band structures of 122-type FeSCs, we performed density functional theory (DFT) to calculate the electronic band structures for the following systems: one unit-cell layer of Ba$_{0.55}$K$_{0.45}$Fe$_2$As$_2$ (BKFA45)

with an As-terminated surface exposed, KFe$_2$As$_2$ (KFA), one unit-cell layer of Ba$_{0.23}$K$_{0.77}$Fe$_2$As$_2$ (BKFA77) with an As-terminated surface exposed, and one unit-cell layer of KFA with an As-terminated surface exposed (details of the simulation can be found in the Methods section). As shown in Fig. 4a, the calculated electronic band structure of KFA consists of saddle points situated at the midpoint V$_0$ of $\Gamma - M$ (indicated by a purple triangle). Considering that surface hole doping only influences the electronic structure of the surface layer, here we focus exclusively on the calculated band structures of the single layers of Ba$_{1-x}$K$_x$Fe$_2$As$_2$ with an exposed As-terminated surface. Our calculation results (Fig. 4a) indicate that with increased hole doping ($x$), the

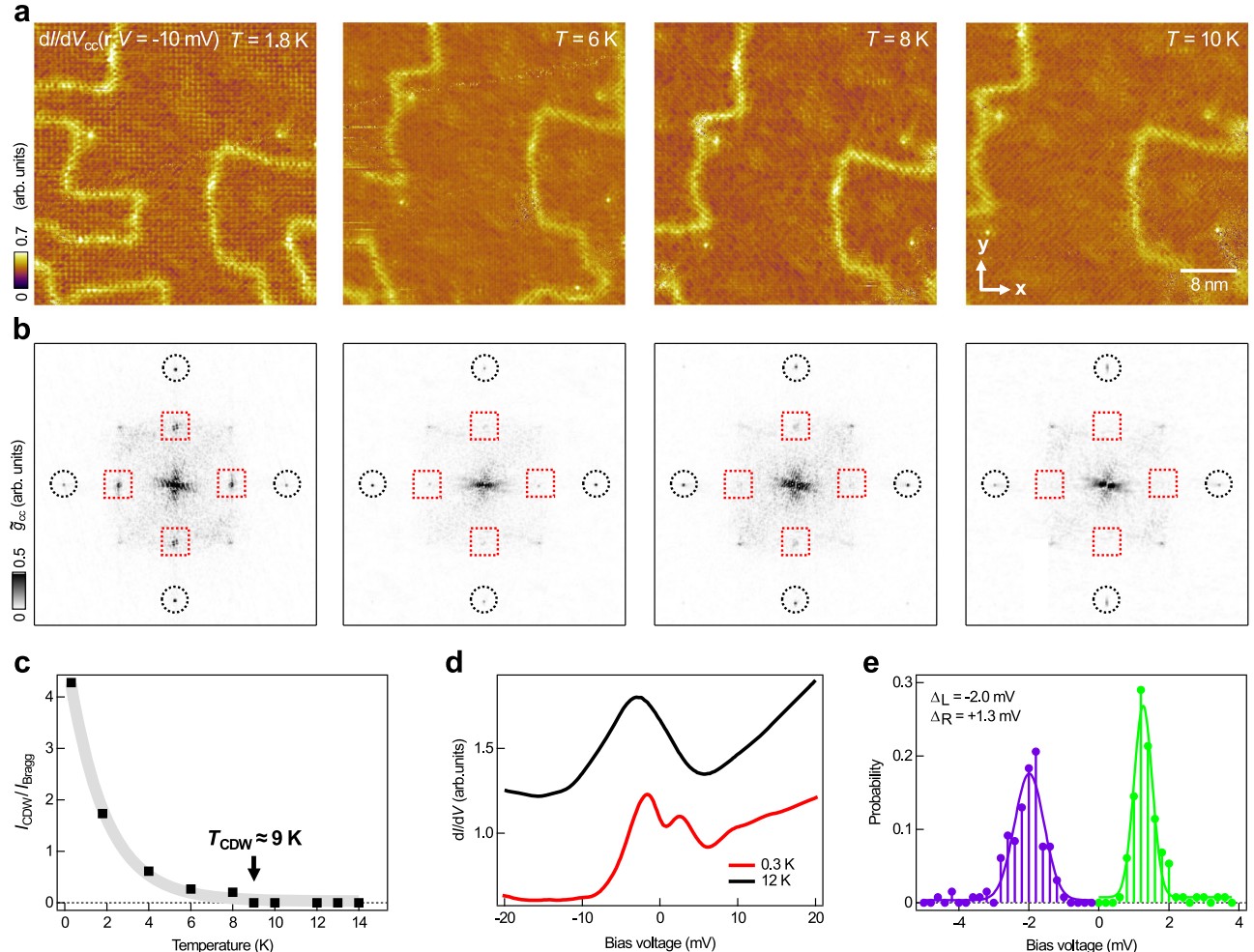

**Fig. 3 | Temperature dependent evolution of the (2 × 2) CDW in Ba$_{0.23}$K$_{0.77}$Fe$_2$As$_2$.** **a** Temperature-dependent constant current differential conductance maps d$I$/d$V_{cc}$(**r**, $V$) at $V$ = −10 mV reveals the progressive attenuation of the CDW, which becomes completely suppressed above 10 K [($V_s$, $I_s$) = −10 mV, 200 pA; $V_{mod}$ = 0.75 mV]. All map slices were taken in the same surface location. **b** The associated Fourier transformations of **a**. Black circles mark the As- lattice peaks, red squares mark the charge order peaks. **c** Plot of the extracted intensities of the CDW peak as a function of temperature. All the peak intensities are normalised to that of the As- unit cell Bragg peak. The grey solid line shows the trend of the intensity drop. **d** Point d$I$/d$V$ spectra taken from the As-terminated surface at two different temperatures: 300 mK and 12 K [($V_s$, $I_s$) = − 10 mV, 200 pA; $V_{mod}$ = 0.25 mV]. The spectra are vertically offset for clarity. **e** Histogram of the spatial distribution of the CDW peak energies. Solid purple and green lines are the results from numerical fittings using a Gaussian function.

electronic band structure of Ba$_{1-x}$K$_x$Fe$_2$As$_2$ with an As-terminated surface has the saddle points moving closer to $E_F$. In KFe$_2$As$_2$, the saddle points are located just a few meV below $E_F$, leading to a strong and sharp DOS peak in d$I$/d$V$ spectra[16]. The tunnelling spectra measured on the As-terminated surfaces of BKFA77 and KFA also exhibit a strong DOS peak near the Fermi level. Both the As-terminated surfaces of BKFA77 and KFA exhibit higher levels of hole doping compared to KFe$_2$As$_2$. As a result, on the As-terminated surface of BKFA77 the DOS peak is exactly at $E_F$, while on that of KFA, the DOS peak is located at 3 meV above the Fermi level (Fig. 4b). This observation is highly consistent with the predicted levels of hole doping, indicating that the DOS peak originates from the saddle points. However, the d$I$/d$V$ spectra measured on the As-terminated surface of BKFA45 exhibits a superconducting gap without such a DOS peak (Supplementary Fig. S10), whose absence may be attributed to the saddle points being far from $E_F$ and buried within other energy bands. As such, our calculation and STM/S results altogether show that the surface doping effect in BKFA77 and KFA results in higher levels of hole doping on the top surface layers compared to KFe$_2$As$_2$, and as a consequence, the Van Hove singularity moves towards $E_F$.

Combining the previous ARPES measurement[16] with our STM/S results, the near $E_F$ band structure for this particular scenario is depicted in Fig. 4c, showing the two-dimensional saddle points located near $E_F$ at the midpoint of the principal axes within the first Brillouin zone. When multiple saddle points are present near $E_F$, the combination of a divergent density of states (DOS) with a near-nested Fermi surface strongly amplifies electron-electron interactions, resulting in various ordering instabilities[43-51]. A prominent example is CDW. As predicted by Rice and Scott[43], the electronic susceptibility $\chi_0(\mathbf{q})$ of a two-dimensional system diverges logarithmically at a wavevector $\mathbf{q_c}$ that connects two saddle points. This suggests a "new scenario" for the emergence of CDW instabilities, one that deviates from the conventional Fermi surface nesting or electron-phonon coupling scenario. Based on this scenario, we calculated the electronic susceptibility $\chi_0(\mathbf{q})$ of Ba$_{0.23}$K$_{0.77}$Fe$_2$As$_2$ (Fig. 4d) (see Methods for calculation details). The wavevector $\mathbf{q_c}$ of the strong peaks in the calculated $\chi_0(\mathbf{q})$ matches with the nesting vector between the near-$E_F$ saddle points, which itself is equal to the wavevector of the CDW ($q_{CDW} = \pi/a_{As}$) as observed in STM. The differential conductance signal exhibits a minor dip at $E_F$ (Fig. 4b), indicating the removal of a portion of the DOS at the saddle points following the CDW formation. In addition, the ratio $\Delta_{CDW}/k_B T_{CDW}$ measured by experiment is about 2.12, in good agreement with the theoretical prediction[43] (the averaged gap

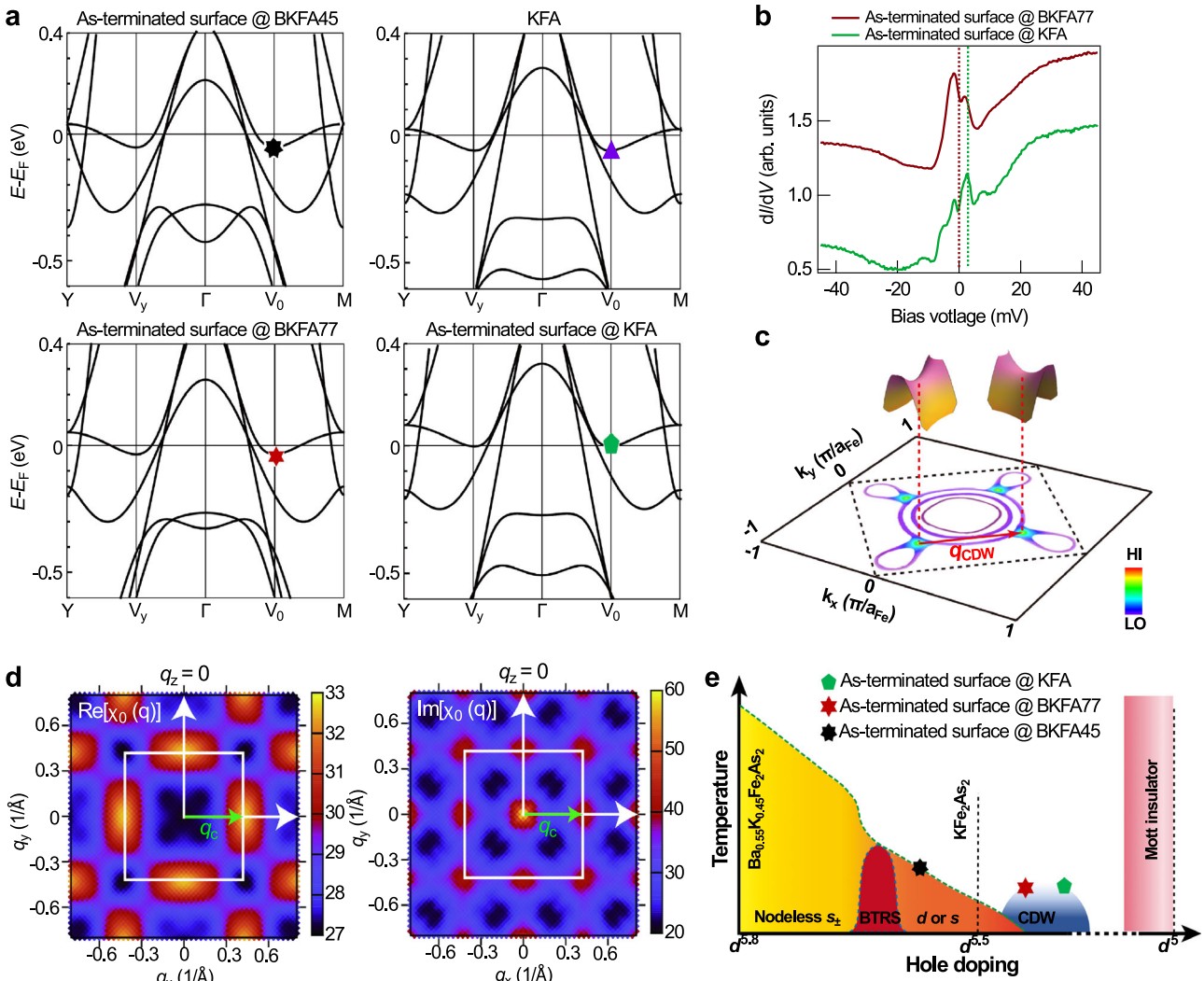

**Fig. 4 | Saddle points near the Fermi level. a** Calculated band structures of (top-left) one unit-cell layer of $Ba_{0.55}K_{0.45}Fe_2As_2$(BKFA45) with an As-terminated surface exposed, (top-right) $KFe_2As_2$(KFA), (bottom-left) one unit-cell layer of $Ba_{0.23}K_{0.77}Fe_2As_2$(BKFA77) with an As-terminated surface exposed, and (bottom-right) one unit-cell layer of KFA with an As-terminated surface exposed. Coloured markers indicate the positions of the saddle points. **b** Large range $dI/dV$ spectra taken from the As-terminated surfaces of BKFA77 [$(V_s, I_s)$ = 90 mV, 1 nA; $V_{mod}$ = 0.5 mV] and KFA [$(V_s, I_s)$ = 80 mV, 500 pA; $V_{mod}$ = 0.5 mV]. The spectra are vertically offset for clarity. **c** Schematic diagram showing the proposed near Fermi-energy band structure of the As-terminated surface of a heavily-hole-doped iron arsenide superconductor as a result of surface doping. The saddle points are located at the midpoints between $\Gamma - M$. The vector that connects any two adjacent saddle points is consistent with the wavevector of the 2 × 2 CDW. Density of states plot within the first Brillouin zone is based on the previous work[16]. **d** The electronic

susceptibility $\chi_0(\mathbf{q})$ plot. **e** Schematic showing the extended phase diagram of 122-type iron-based superconductors. At light hole-doping, the superconducting pairing state is most likely $s\pm$. A narrow dome with broken time-reversal symmetry (BTRS) exists inside the broad $s$-wave superconductivity dome of the $Ba_{1-x}K_xFe_2As_2$ phase diagram. The pairing order parameter near $KFe_2As_2$ remains elusive, with both $d$- and $s$- waves being most favourable. A Mott insulating phase is strongly favoured by Hund's coupling at the $3d^5$ configuration. The blue area denotes the CDW phase beyond the $3d^{5.5}$ configuration. The hole-doping level of BKFA77 with an As-terminated surface is indicated by a red hexagon, while that of KFA with As-terminated surface by a green pentagon. The black heptagon represents the hole-doping level of BKFA45 with an As-terminated surface, which exhibits superconductivity with no sign of charge order. Panel **c** adapted with permissions from ref. 16, American Physical Society. https://doi.org/10.1103/PhysRevB.92.144513.

value $\Delta_{CDW}$ is fitted by using Gaussian distribution in Fig. 3d). Both theoretical and experimental results collectively provide strong evidence for the existence of a static charge order that is most likely driven by saddle-point nesting. Recent studies on kagome superconductors $AV_3Sb_5$ (A = K, Rb, or Cs) suggested that the CDW observed in those systems is also driven by saddle-point nesting[52]. However, unlike the two-dimensional saddle points in heavily hole-doped 122-type FeSCs, the saddle points in $AV_3Sb_5$ still exhibit $k_z$ dependence[53]. While the CDW order and the van Hove singularities are closely related, the roles of other fundamental properties, such as electron-phonon coupling[54] and orbital hybridization[55], in these symmetry-breaking phenomena remain largely unclear.

## Discussion

To summarise our interpretation, we propose an extended phase diagram of 122-type iron pnictides as shown in Fig. 4e. In the heavily hole-doped regime (with electron density less than 5.5 electrons/Fe), a Mott insulating state that resembles the parent compound of cuprates was predicted for the $d^5$ configuration. This state influences a large part of the phase diagram. CDW occurs via electron-doping a Mott insulator, as shown in the blue area. The red hexagon indicates the location of $Ba_{0.23}K_{0.77}Fe_2As_2$ with an As-terminated surface, and the green pentagon denotes $KFe_2As_2$ with an As-terminated surface. Further electron doping, as in the case of $Ba_{0.55}K_{0.45}Fe_2As_2$ with an As-terminated surface (indicated by the black heptagon), results in the

suppression of CDW, followed by the emergence of superconductivity. Our observation here therefore expands the phase diagram of FeSCs and offers important clues on the pairing order parameter in $KFe_2As_2$, suggesting that the superconductivity observed in $KFe_2As_2$ may arise from a CDW order. Meanwhile, our findings show that further increasing the hole doping level via surface doping offers a promising platform for investigating correlated electronic states in an extended range of hole doping levels. Moreover, recent $\mu$SR and thermoelectric measurements indicate the breaking of time-reversal symmetry in $Ba_{1-x}K_xFe_2As_2$ ($x \approx 0.77$)[41,42,56]. This altogether raises the possibility for studying the as-yet-discovered interplay between the superconducting order parameter, which breaks time-reversal symmetry in the bulk, and the CDW present at the surface.

## Methods

### Materials
Phase purity and crystalline quality of the single crystals of $Ba_{1-x}K_xFe_2As_2$ ($x \approx 0.77$) were examined by X-ray diffraction (XRD) and transmission electron microscopy (TEM). The $c$-axis lattice parameters were calculated from the XRD data using the Nelson-Riley function. The K doping level $x$ of the single crystals was determined using the relation between the $c$-axis lattice parameter and the K doping obtained in previous studies[57].

### Scanning tunnelling microscopy/spectroscopy
The STM/S experiments were performed using a commercial Unisoku USM1300 low-temperature STM machine that operates at a base temperature of 300 mK. Pt/Ir tips were used and conditioned by field emission on a gold target. To obtain clean surfaces for STM measurements, $Ba_{1-x}K_xFe_2As_2$ samples were cleaved in-situ at ~78 K in ultra-high vacuum (base pressure ~$2 \times 10^{-10}$ mbar), then immediately transferred to the STM stage (maintained at 4.2 K) for STM/S measurements.

Conventional (constant height) differential conductance ($dI/dV$) spectra and maps $dI/dV(\mathbf{r}, V)$ were recorded using a standard lock-in technique, with the frequency of bias modulation set at 973 Hz. Constant current differential conductance map $dI/dV_{cc}(\mathbf{r}, V)$ slices were obtained through continuous acquisition of $dI/dV$ signals for every position of the scanned area in a STM scan at fixed scan set-points ($V_0, I_0$), during which the current feedback loop was kept closed. This method allows for acquisition of differential conductance map slices in a relatively short time, thereby minimizing the effect of thermal drift that typically occurs during prolonged spectroscopic map measurements. Acquisition of each $dI/dV_{cc}(\mathbf{r}, V)$ map slice shown in Fig. 3 took about two hours. Supplementary Fig. S5 shows that $dI/dV_{cc}(\mathbf{r}, V)$ map slices in Fig. 3 do not contain any artifacts, therefore reveal the same features as $dI/dV(\mathbf{r}, V)$ map slices obtained by conventional means. Regarding data processing, all topographic STM images $z(\mathbf{r}, V)$ shown in the manuscript were background subtracted. Point $dI/dV$ spectra were plotted without processing, and vertically offset for clarity when displayed in series. To minimize possible set-point effect, differential conductance as a function of position and energy maps shown in Fig. 2 were presented in the form of normalised differential conductance $L(\mathbf{r}, V) = dI/dV(\mathbf{r}, V)/(I(\mathbf{r}, V)/V)$. Constant-current differential conductance maps $dI/dV_{cc}(\mathbf{r}, V)$ shown in Fig. 3 were presented without any processing.

### Computational details
Density functional theory (DFT) calculations were performed using the full potential local orbital (FPLO) basis[58] and the generalized gradient approximation (GGA) to the exchange and correlation functionals[59]. Interpolations of lattice parameters[60] were used and atomic positions optimized within the GGA. Charge-doping in $Ba_{0.55}K_{0.45}Fe_2As_2$ and $Ba_{0.23}K_{0.77}Fe_2As_2$ was modelled via the virtual crystal approximation at the Ba/K site. On the other hand, the surface state band structures of $Ba_{0.55}K_{0.45}Fe_2As_2$, $Ba_{0.23}K_{0.77}Fe_2As_2$ and $KFe_2As_2$ is estimated by means

of rigid-band approximation. To understand the formation of the CDW at the As-terminated surface, we focused on the Fermi surfaces in the $x$-$y$ plane when $E_F = E_{saddle}$. We calculated the electronic susceptibility $\chi_0(\mathbf{q})$ in $Ba_{0.23}K_{0.77}Fe_2As_2$. Specifically, when the real part of $\chi_0(\mathbf{q})$ diverges, the structure is unstable against the CDW formation. The total electronic susceptibility, that is, the real part of $\chi_0(\mathbf{q})$, is calculated via:

$$\text{Re}[\chi_0(\mathbf{q})] = \frac{1}{N} \sum_{\mathbf{k}, m, n} \frac{n_F(\epsilon_{\mathbf{k+q}}, m) - n_F(\epsilon_{\mathbf{k}}, n)}{\epsilon_{\mathbf{k}, n} - \epsilon_{\mathbf{k+q}, m}} \quad (1)$$

where $N$ is the number of the unit cell, $\epsilon_{\mathbf{k}, m}$ the eigen-energy for the $m^{\text{th}}$ band at momentum $\mathbf{k}$, and $n_F(\epsilon) = 1/[\exp(\epsilon/k_B T) + 1]$ denotes the Fermi-Dirac distribution function. The imaginary part of $\chi_0(\mathbf{q})$ is written as

$$\text{Im}[\chi_0(\mathbf{q})] = \frac{1}{N} \sum_{\mathbf{k}, m, n} \delta(\epsilon_{\mathbf{k}}, n)\delta(\epsilon_{\mathbf{k+q}}, m) \quad (2)$$

## Data availability
Source data in this study are provided in the Supplementary Information/Source Data file. Other related data are available from the corresponding author(s) upon request. Source data are provided with this paper.

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

## Acknowledgements

We thank Hu Miao, Lingyuan Kong, Noah F. Q. Yuan, and Egor Babaev for helpful discussions. C.M.Y. acknowledges support from the Ministry of Science and Technology of China (2022YFA1402702), Shanghai Pujiang Talent Program (No. 21PJ1405400), TDLI Start-up Fund. H.D.

acknowledges support from the New Cornerstone Science Foundation (No. 23H010801236), Innovation Program for Quantum Science and Technology (No. 2021ZD0302700). B.L. acknowledges support from Natural Science Foundation of China (Grant 12374063), the Ministry of Science and Technology of China (2023YFA1407400), the Shanghai Natural Science Fund for Original Exploration Program (23ZR1479900), and the Cultivation Project of Shanghai Research Center for Quantum Sciences (Grant No. LZPY2024). V.G. acknowledges support from Natural Science Foundation of China (Grants 12374139 and 12350610235). Q.H. acknowledges support from China Postdoctoral Science Foundation (No. GZB20230421).

## Author contributions

H.D., Q.H., C.M.Y. and B.L. designed the experiment. Q.H., Y.Z. and F.Y. performed the STM measurements. Q.H. and Z.Y. analysed the data. H.X., J.D. and C.L. and Z.W. performed DFT calculations. V.G. provided $Ba_{1-x}K_xFe_2As_2$ single-crystal samples. Q.H., Y.Z., B.L., C.M.Y. and H.D. wrote the manuscript with input from all other authors. B.L., H.D. and C.M.Y. supervised the project.

## Competing interests

The authors declare no competing interests.
