## [Transparent Peer Review file · Nature Communications]

Evidence for saddle point-driven charge density wave on the surface of heavily hole-doped iron arsenide superconductors

Corresponding Author: Professor Chi Ming Yim

Version 0:

Reviewer comments:

Reviewer #1

(Remarks to the Author)

The authors have addressed my major concerns. It is a nice manuscript and I am happy to support publication in Nature Communications.

Reviewer #2

(Remarks to the Author)

All questions have been addressed thoroughly by the authors. I appreciate the extra effort to provide additional data. From my side, the only point remaining is the question of how to present the hypothesis of a saddle-point driven CDW. The evidence remains circumstantial.

- The new data in Fig. 4b does not show a shift of the peak between $x=0.77$ and $x=1$. The dashed lines are misleading in my opinion. If the authors plot both curves on top of each other, the only difference is in the fine structures and relative intensities, the overall peak does not shift.

- I could only find data on $x=0.45$ inside the superconducting state (Fig S10) and the question whether a vHS exists close to EF can therefore not clearly be answered.

- The authors wrote that other experimental evidence in literature for where the vHS is located in the (K,Rb,Cs)₁₂₂ series provides conflicting information.

- Correlation effects are not included in DFT and therefore the calculations can only serve as an initial guess.

All in all, I suggest to tone down the claim for a saddle-point driven CDW and present it as a hypothesis as was suggested in the previous round.

We thank the reviewers for spending their invaluable time evaluating our manuscript. Below, we will address their comments/questions on a point-to-point basis.

REVIEWERS' COMMENTS

Reviewer #1 (Remarks to the Author):

The authors have addressed my major concerns. It is a nice manuscript and I am happy to support publication in Nature Communications.

Reply: We thank Reviewer 1 for his/her recognition on our work.

Reviewer #2 (Remarks to the Author):

All questions have been addressed thoroughly by the authors. I appreciate the extra effort to provide additional data. From my side, the only point remaining is the question of how to present the hypothesis of a saddle-point driven CDW. The evidence remains circumstantial.

- The new data in Fig. 4b does not show a shift of the peak between $x=0.77$ and $x=1$. The dashed lines are misleading in my opinion. If the authors plot both curves on top of each other, the only difference is in the fine structures and relative intensities, the overall peak does not shift.

Reply: We agree with the reviewer that the shift of the vHs peak between the curves in Fig. 4b might not be as obvious as we stated, especially when there are also a number of fine structures present near the Fermi level (E_F). To further justify the peak shift, here we show a comparison of the spectra of obtained from the As-terminated surfaces of samples of different x values at a temperature of 12 K, at which all superconductivity and charge order (etc) related features should already disappear [here we also note that the 12 K spectra obtained from the As-terminated surfaces of $x = 0.77$ and $x = 1$ samples have been provided in Fig. 3c (black curve) and Fig. S9 (black curve) respectively in the last submission package]. From Figure R1, one can see that both spectra exhibit a peak (which largely corresponds to the vHs) around E_F , with that in the spectrum obtained from the $x = 0.77$ sample located at about -2 meV below E_F while that in the spectrum from the surface of the $x = 1$ sample at about +1 meV above E_F . This serves as solid evidence for the upward shift of the vHs peak in STS with increasing x . We have included Fig. R1 as Fig. S12 in the revised Supplementary Information.

Figure R1: 12 K differential conductance dI/dV spectra taken from the As- terminated surfaces of BKFA samples with two different levels of K-doping: (black) $x = 0.77$, (red) $x = 1$. Spectroscopic set-points: $(V_s, I_s) = (-10 \text{ mV}, 200 \text{ pA})$; $V_{\text{mod}} = 0.25 \text{ mV}$ for the black curve, and $[(V_s, I_s) = (-20 \text{ mV}, 400 \text{ pA})$; $V_{\text{mod}} = 0.25 \text{ mV}$] for the red curve.

- I could only find data on $x=0.45$ inside the superconducting state (Fig S10) and the question whether a vHS exists close to EF can therefore not clearly be answered.

Reply: We understand the reviewer's concern, but unfortunately, we did not succeed in obtaining a normal state spectrum (12 K) from the As- terminated surface of the $x = 0.45$ sample.

Back to the superconducting state spectrum (see Fig. R2), the vHS peak, which should be located somewhere between -5 and -10 mV for the As-terminated surface of the $x = 0.45$ sample, does not unambiguously show up in the spectrum. We attribute its absence in the spectrum to the saddle points being too far from the Fermi level and buried within other energy bands within the band structure (also see QPI map Fig. S7h).

Figure R2: Superconducting state dI/dV spectrum of the As-terminated surface of $\text{Ba}_{0.55}\text{K}_{0.45}\text{Fe}_2\text{As}_2$ sample. The spectrum shown was extracted from the average of a 1D spectroscopic map obtained at a temperature of 300 mK. Spectroscopy measurement parameters: $(V_s, I_s) = (-30 \text{ mV}, 500 \text{ pA})$, $V_{\text{mod}} = 0.5 \text{ mV}$.

- The authors wrote that other experimental evidence in literature for where the vHS is located in the (K,Rb,Cs)122 series provides conflicting information.

Reply: We thank the reviewer for raising this viewpoint again which we now agree about. On this basis, we have removed the entire related paragraph in the Discussion section in the revised manuscript.

- Correlation effects are not included in DFT and therefore the calculations can only serve as an initial guess.

Reply: Based on the reviewer's comment, we have rephrased the related sentence in the calculation part in the main text (page 8) as below:

"To understand qualitatively the influence of surface hole-doping on the electronic band structures of 122-type FeSCs..." (page 8)

All in all, I suggest to tone down the claim for a saddle-point driven CDW and present it as a hypothesis as was suggested in the previous round.

Reply: Based on the reviewer's comment, We have rephrased the related sentence in the abstract as follow:

"Notably, the CDW emerges when the saddle points approach the Fermi level, where its wavevector matches with those linking the saddle points, suggesting saddle-point nesting as its most probable formation mechanism."